# Extraction of Seed Oil from *Heracleum persicum* Desf. ex Fischer and Investigation of Its Composition, Qualitative and Nutraceutical Properties

**DOI:** 10.3390/foods14203486

**Published:** 2025-10-13

**Authors:** Abdolah Dadazadeh, Sodeif Azadmard-Damirchi, Zahra Piravi-Vanak, Mohammadali Torbati, Fleming Martinez

**Affiliations:** 1Department of Food Science and Technology, Faculty of Agriculture, University of Tabriz, Tabriz 51666, Iran; a.dadazadeh@tabrizu.ac.ir; 2Food, Halal, and Agricultural Products Research Group, Food Technology and Agricultural Products, Research Center, Standard Research Institute (SRI), Karaj 31745, Iran; 3Department of Food Science and Technology, Faculty of Nutrition, Tabriz University of Medical Sciences, Tabriz 15731, Iran; torbatim@tbzmed.ac.ir; 4Pharmaceutical-Physicochemical Research Group, Department of Pharmacy, Faculty of Science, The National University of Colombia, Bogotá 11001, Colombia; fmartinezr@unal.edu.co

**Keywords:** nutritional indices, alpha-tocopherol, medicinal plants, traditional Iranian medicine, oilseeds

## Abstract

*Heracleum persicum* Desf. ex Fischer, a species of the Apiaceae family, is endemic to Iran and has been historically utilized as a spice, condiment, and medicinal plant. The plant produces seeds that represent a potential new source of vegetable oil. In this study, the oil from these seeds was extracted using a solvent, and its physical, chemical, and nutritional properties were investigated. The oil extraction yield was determined to be 12.62%. Oleic acid (61.11%) and linoleic acid (25.84%) were identified as the predominant fatty acids in the extracted oil. Among its phytosterols, beta-sitosterol (65.6%) and stigmasterol (14.0%) were the most abundant. Furthermore, this oil exclusively contained alpha-tocopherol at a relatively high concentration (1610.9 ppm). The chlorophyll and carotenoid contents of the extracted oil were 28.34 mg/kg and 4.95 mg/kg, respectively. Regarding its nutritional indices, the atherogenic index, thrombogenic index, and hypocholesterolemic to hypercholesterolemic ratio were 0.13, 0.24, and 9.77, respectively. In conclusion, considering its unique oil composition and qualitative characteristics, this oil holds promise as a novel source of vegetable oil and a valuable byproduct of *Heracleum persicum*.

## 1. Introduction

*Heracleum persicum* (HP) is a perennial flowering plant belonging to the Apiaceae family. It typically grows to a height of 1.5 to 2 m, with a hollow, reddish-brown base (15–20 mm in diameter) [1]. The stems are notably hollow and thick, while the leaves are dark green, possess a large base, and are glabrous on the surface, but hairy on the underside. Stem leaves gradually decrease in size towards the top of the plant. The umbels are large, ranging from 30 to 50 radially arranged, with unequal sizes, and are slightly or sparsely glabrous. The petals are white, and the seeds are oval and very thin (Figure 1). Ten *Heracleum* species grow wild in Iran, with four of them being endemic to the region [2].

Several studies have been conducted on secondary metabolites of HP, especially on its essential oil and extracts. HP seeds extracted with acetone have been shown to contain alkaloids, triterpenoid saponins, and flavonoids [3]. A laboratory study conducted in an animal model demonstrated that the essential oil from HP seeds, acting as a potent herbal medicine, can function as an antioxidant to inhibit oxidative stress and protect the liver from significant damage [4].

This plant exhibits various phytopharmacological effects. For instance, in Europe, it is used for the treatment of herpes zoster, colds, and rheumatism [5]. In traditional and complementary medicine, HP is also employed as an analgesic, anticonvulsant, and disinfectant, and is additionally utilized as a flavoring spice in the preparation of pickles [6]. The essential oil derived from this plant possesses antispasmodic, antioxidant, and anti-inflammatory properties [1,7]. Furthermore, it has been reported that it can exhibit antimicrobial properties, proving effective against bacteria and fungi such as *Candida utilis* and *Candida albicans*. The combination of HP essential oil with chitosan has also been shown to extend the shelf life of marine products and prevent fish spoilage during storage and warehousing. Recent research has further proposed the anti-inflammatory, analgesic, and immune system-modulating properties of this plant [1].

However, despite these studies, very little is known about the composition and nutritional properties of HP seed oil. Therefore, this study was conducted to investigate and determine these properties of HP seed oil. Identifying new oil sources offers numerous benefits, including reducing human dietary dependence on existing oil sources. Moreover, a newly introduced oil source can find diverse applications in the pharmaceutical, cosmetic, and food industries.

## 2. Materials and Methods

### 2.1. Materials

HP seeds were sourced from a local market. Chemicals utilized in this study, including acetic acid, chloroform, *n*-hexane, phenolphthalein, sodium hydroxide, cyclohexane, sodium thiosulfate, potassium hydroxide were from Merck, Darmstadt, Germany, and Folin-Ciocalteu reagent and starch were from Sigma-Aldrich, St. Louis, MO, USA. Distilled water was prepared using a German GFL (Gesellschaft für Labortechnik, Berlin, Germany) water distillation device.

### 2.2. Moisture Content Determination

The moisture content of HP seeds was determined according to the method described by the American Chemical Society [8]. It was calculated using the following formula:Moisture (%) = (W_1_ − W_2_)/W_1_ × 100

W_1_ = Seed weight (in grams) before drying

W_2_ = Seed weight (in grams) after drying

### 2.3. Seed Oil Extraction

The powdered HP seeds were poured into a glass container with a lid, then *n*-hexane was used to extract the oil. Solvent to seed powder ratio was 10 to 1 (*v*/*w*). Oil extraction was performed on a shaker with continuous agitation for 6 h at 25 °C. Subsequently, the solid meal was separated from the solvent-oil mixture. The solvent was then evaporated under vacuum using a rotary evaporator to obtain the pure oil [3].

### 2.4. Oil Extraction Yield Calculation

The oil yield of HP seeds was calculated using the following formula:Oil Yield (%) = (M_1_/M_2_) × 100

M_1_ = Amount of oil obtained from seeds (grams)

M_2_ = Weight of seeds used in oil extraction (grams)

### 2.5. Specific Gravity Measurement

A pycnometer was first weighed after drying in an oven and cooling in a desiccator (W_1_). It was then filled with distilled water and weighed (W_2_). Subsequently, the same pycnometer was filled with HP oil and weighed (W_3_). The measurements were done at room condition. The specific gravity was calculated using the following formula [9]:$$Specific\;gravity=\frac{\mathrm{weight}\mathrm{of}oil\left(W_{3}-W_{1}\right)}{\mathrm{weight}\mathrm{of}\mathrm{equal}\mathrm{volume}\mathrm{of}water\left(W_{2}-W_{1}\right)}$$$$Density\left(g/mL\right)=\frac{\mathrm{weight}\mathrm{of}oil}{\mathrm{weight}\mathrm{of}water}$$

### 2.6. Refractive Index Measurement

The refractive index of the extracted oil was measured at 25 °C using the AOCS method Cc 7-25, as described by the American Oil Chemists’ Society [9].

### 2.7. Acid Value Determination

The acid value (AV) of the obtained oil was determined according to the AOCS method Cd 3d-63, as recommended by AOAC [6].

### 2.8. Peroxide Value Determination

The peroxide value of the extracted oil was measured following the ISO 3976 (IDF 74:2006) method, as detailed by the Dairy Federation [10].

### 2.9. Carotenoid Content Measurement

After preparing the oil solution with cyclohexane, the carotenoid content was determined spectrophotometrically (UNICO UV/Vis 2100, Franksville, WI, USA) at a wavelength of 450 nm [7]. The content was calculated using the following formula:$$Carotenoid\left(mg/kg\right)=\frac{\left(A450\times 10^{6}\right)}{\left(2000\times 100\times d\right)}$$

A450 = absorbance at 450 nm

*d* = spectrophotometer cell thickness (1 cm)

### 2.10. Chlorophyll Content Measurement

After preparing the oil solution with cyclohexane, the chlorophyll content was determined spectrophotometrically at a wavelength of 670 nm [4]. The content was calculated using the following formula:$$Chlorophyll\left(mg/kg\right)=\frac{\left(A670\times 10^{6}\right)}{\left(613\times 100\times d\right)}$$

A670 = absorbance at 670 nm

*d* = spectrophotometer cell thickness (1 cm)

### 2.11. Total Phenolic Compounds Determination

The total phenolic compounds in the oil were quantified using the method described by Caponio et al. [11] expressed as mg gallic acid equivalents per kg of oil sample. This was performed spectrophotometrically at a wavelength of 765 nm using Folin-Ciocalteu reagent.

### 2.12. Fatty Acid Profile Analysis

For the quantitative and qualitative analysis of fatty acid profiles, a GC (Agilent Technologies, Santa Clara, CA, USA) equipped with a BPX70 capillary column (50 m length, 0.25 mm internal diameter, and 0.25 µm stationary phase particle size) and a flame ionization detector (FID) was utilized. Helium and nitrogen were used as the carrier gas and as the make-up gas, respectively. Temperature and time programming were performed according to the method described by Dadazadeh et al. [12].

### 2.13. Triacyclglycerols Analysis

Analysis of triacylglycerols was conducted following the methodology outlined by Farmani et al. (2006) [13]. Utilizing a GC system (Agilent 7890 B, Santa Clara, CA, USA), and oven temperature of 350 °C, the obtained individual peaks were identified by comparing their retention times against those of established triacylglycerol standards.

### 2.14. Phytosterols Analysis

Phytosterol compounds were detected and quantified using an Agilent Technologies GC device. Initially, the oil sample was saponified with ethanolic potassium hydroxide (2 mol/L), and the unsaponifiable fraction containing phytosterols was separated. Subsequently, derivatization was performed with trimethylsilyl (TMS) reagent. The GC analysis was conducted using a fused silica column (30 m length, 0.18 mm diameter, and 0.18 µm stationary phase particles). Helium was employed as the carrier gas and nitrogen as the make-up gas. The detector was a flame ionization detector (FID). Temperature and time programming followed the method established by Dadazadeh et al. (2025) [12]. Phytosterols in the tested oil were identified based on their retention times, and their quantities were determined using 5α-cholestane as an internal standard [14].

### 2.15. Measurement of Tocopherols

The tocopherol content in the oil was measured using a HPLC system, following the method described by Azadmard Damirchi and Dutta (2008) [12]. Tocopherols were identified using a Reverse Phase HPLC (YOUNGLIN Acme 9000, Kungsbacka, Sweden) equipped with a Teknokroma BRISA LC2 C18 column (4.6 mm × 150 mm, 5 µm particle size) (Barcelona, Spain). The mobile phase consisted of 98% methanol and 2% water, with a flow rate of 1 mL/min. A UV detector set at a wavelength of 250 nm was used for detection.

### 2.16. Nutritional Indices and Nutraceutical Properties

Given the well-established effects of saturated and unsaturated fatty acids on the risk of cardiovascular diseases, Ulbricht and Southgate (1991) proposed specific ratios, namely the Index of Atherogenicity (AI) and the Thrombogenicity Index (TI), to evaluate the nutritional quality of fats and oils [15]. These formulas were later refined and modified [16].

Other nutritional indices, such as the hypocholesterolemic to hypercholesterolemic ratio (hypo/hyper index), were subsequently calculated and determined [10]. These formulas, detailed below, were employed to assess the nutritional properties and indices of HP oil:$$\mathrm{Atherogenicity}Index=\frac{C12:0+4\times C14:0+C16:0}{\Sigma MUFA+\Sigma (\omega 6)+\Sigma (\omega 3)}$$$$\mathrm{Thrombogenicity}Index=\frac{C14:0+C16:0+C18:0}{0.5\times \Sigma MUFA+0.5\times \Sigma (\omega 6)+\Sigma (\omega 3)}$$$$Hypo/\mathrm{Hyper}Index=\frac{C18:1+C18:2+C18:3+C18:4+C20:4}{C14:0+C16:0}$$

### 2.17. Statistical Analysis

All experiments were performed in triplicate, and the statistical analysis of the collected data was calculated as mean ± standard deviation (SD) using Microsoft Excel 2019.

## 3. Results and Discussion

### 3.1. Physical and Chemical Properties of HP Seed Oil

The physical and chemical properties of *Heracleum persicum* seed oil were measured and are presented in Table 1.

#### 3.1.1. Oil Percentage

The oil content in oilseeds holds significant economic importance. *Heracleum persicum* seeds were found to contain 12.62% oil. This value is comparable to that of grape seed oil (5.40–10.79%) [17] and pomegranate seed oil (10.8–15%) [3,18], but it is lower than that reported for soybean oil (18–25%), safflower (27–32%) [19], flaxseed (33–42%) [20], and canola (35–40%) [21].

#### 3.1.2. Specific Gravity

Specific gravity is an important physical characteristic of oil, playing a significant role in the oil identification and the detection of adulteration [22]. The specific gravity of HP oil was 0.876 at 20 °C. This value is generally lower than that of most common vegetable oils, including sesame (0.915–0.924), flax seed (0.925–0.935), sunflower (0.916–0.923), soybean (0.919–0.925), corn (0.917–0.925), safflower (0.913–0.919), rapeseed (0.910–0.920), mustard (0.910–0.921), coconut (0.908–0.921), pistachio (0.915–0.920), almond (0.911–0.929), and hazelnut (0.898–0.915). However, it shows a slight similarity to palm oil (0.866–0.899) [23].

#### 3.1.3. Refractive Index

The refractive index is another key physical property that aids in oil identification and purity assessment, alongside with other quality control factors. The refractive index of HP oil was 1.464. This value is lower than that reported for almond oil (1.468–1.475), flaxseed (1.472–1.487), grape seed (1.467–1.477), hazelnut (1.468–1.473), and pistachio (1.467–1.470). Conversely, it is comparable to corn oil (1.465–1.468), sunflower (1.461–1.475), rice bran (1.460–1.473), sesame (1.465–1.469), rapeseed (1.465–1.469), mustard (1.461–1.469), and cottonseed (1.458–1.466) [23].

#### 3.1.4. Acid Value

Free fatty acids (FFAs) are generated through the hydrolysis of triglycerides. The concentration of FFAs in an oil serves as a quality indicator, with a generally accepted threshold not exceeding 4 mg KOH/g oil [23]. Elevated FFA levels can compromise oil quality, alter its taste, increase susceptibility to oxidation, and reduce overall stability. Furthermore, consumption of oils with high FFA content has been linked to adverse health effects and increased inflammatory responses in the body [24]. In HP oil, the FFA content was 3.6 mg KOH/g oil, which falls within the permissible range for edible oils as specified by the Codex Alimentarius [23]. A comparative analysis of the acid value of HP oil with other oilseeds indicates that its acid value is higher than that of palm oil (2.64 mg KOH/g oil), sesame (1.34 mg KOH/g oil), mustard oil (0.56 mg KOH/g oil), soybean (0.35 mg KOH/g oil), and coconut (0.48 mg KOH/g oil) [22]. Nonetheless, the acid value of the extracted HP oil remains within the acceptable range of less than 4 mg KOH/g oil [22].

#### 3.1.5. Peroxide Value

Peroxide compounds in oils are formed through the reaction of lipid molecules with oxygen, a process influenced by factors such as temperature, light, air exposure, and enzyme activity [25]. The peroxide value is a critical measure for assessing oil deterioration and oxidative rancidity, reflecting the extent of harmful compound formation [26]. The peroxide value of HP oil was found to be 5.95 (meq O_2_/kg oil), which is within the permissible range of up to 10 (meq O_2_/kg oil) according to the Codex standard [23]. When compared to certain other oils, the peroxide value of HP seed oil is slightly higher than that of soybean oil (4.55 meq O_2_/kg oil) [27] and notably higher than mustard (3.08 meq O_2_/kg oil) [28] and sunflower (3.12 meq O_2_/kg oil) [27] oils. But in general, both the peroxide value and acid vale of HP oil were found to be within acceptable limits for edible purposes according to standard guidelines [23].

#### 3.1.6. Carotenoid Content

Carotenoids are natural pigments found in plants and oilseeds, renowned for their antioxidant properties that inhibit oil oxidation. By neutralizing free radicals, carotenoids delay spoilage and enhance the stability and shelf life of oils [29]. Moreover, these pigments serve as precursors to vitamin A, converting into the vitamin in the body and contributing to skin and eye health [30]. The carotenoid content of HP seed oil was 4.95 (mg/kg). This concentration is lower than that observed in oils such as grape seed (67.99 mg/kg) [31], rapeseed (63.60 mg/kg) [28], sunflower (21.20 mg/kg) [32], and soybean (24.96 mg/kg) [27]. Low carotenoid contents can adversely affect the oxidative stability of this oil, however, as this oil has high percentage of oleic acid (as a monounsaturated fatty acid), which is more stable against oxidation compared to poly unsaturated fatty acids and also high content of alpha-tocopherol (as an antioxidative component), these can compensate the low amount of carotenoids in the oxidative stability of this oil.

It should be noted that carotenoid content can be affected by the varietal differences, seed maturity, extraction method, extraction solvent type and also seed storage conditions. Also, there is a need for further analysis of this oil by the advanced methods such as high-performance liquid chromatography to determine the carotenoid types and the accurate amount of each type.

#### 3.1.7. Chlorophyll Content

Chlorophylls, which are green pigments, can be transferred into the oil during the extraction process from plants and seeds. While visually appealing in some contexts, chlorophylls are known to act as prooxidants, potentially accelerating the oxidative spoilage of oils [33]. To mitigate this effect and enhance oil stability, manufacturers often remove chlorophyll through refining processes or package chlorophyll-rich oils in dark, opaque containers to minimize light exposure [34]. The HP seed oil was found to contain 28.34 ppm of chlorophyll. This level is comparable to that observed in rapeseed oil (26.00 ppm) [31] but is notably higher than in other common vegetable oils such as soybean oil (2.43 ppm), sunflower oil (5.62 ppm) [27,32], and grape seed oil (17.04 ppm) [31].

#### 3.1.8. Phenolic Compounds

Phenolic compounds are a diverse group of phytochemicals widely recognized for their nutritional significance and beneficial effects on human health. These compounds possess strong antioxidant, anti-inflammatory, and anticarcinogenic properties, contributing to the prevention of various chronic diseases, including cancer and inflammatory conditions [34]. In oils, phenolic compounds play a crucial role in enhancing shelf life by inhibiting free radical formation and reducing oxidative reactions, thereby preventing spoilage [35]. The total phenolic content in HP oil was determined to be 1.4 mg/kg oil. This concentration is relatively lower compared to some other oils, but shows similarity to sunflower oil (4.9 mg/kg oil), mustard oil (5.6 mg/kg oil), and sesame oil (3.3 mg/kg) [36]. However, it is considerably lower than the phenolic content found in coconut oil (18 mg/kg oil) and peanut oil (30.9 mg/kg oil) [36].

The phenolic compounds content in this oil was determined by the spectrophotometric method using the Folin-Ciocalteu reagent, which mostly shows the simple phenolic components and cannot determine the complex phenolic components. High performance liquid chromatography is suitable for detailed and accurate determination of this bioactive composition to have better picture of the phenolic composition and content. This lower amount of phenolic compounds of this oil may affect its oxidative stability, however, fatty acid composition and other antioxidative components such as tocopherols can have supporting roles in the oxidative stability as well.

### 3.2. Fatty Acid Profile

The fatty acid composition of an oil significantly influences its technological properties, shelf life, and oxidative stability. Oils rich in saturated fatty acids generally exhibit greater resistance to oxidative degradation [37]. Furthermore, the balance between saturated and unsaturated fatty acids dictates the physical state of the oil (solid vs. liquid) at room temperature; a higher proportion of unsaturated fatty acids results in a more liquid oil, while a greater abundance of saturated fatty acids leads to a more solid consistency [38]. This characteristic is particularly important for various applications within the food industry. Additionally, oils with higher saturated fatty acid content tend to be more heat-resistant and less prone to breaking down into harmful substances during heating and cooking. Conversely, while oils rich in unsaturated fatty acids may be less stable against heat and oxidation, they are nutritionally superior from a health perspective. These fatty acids are crucial for preventing cardiovascular diseases, certain cancers, diabetes, and inflammatory conditions [39].

Analysis of HP seed oil revealed the presence of nine distinct fatty acids, as detailed in Table 2. These include decanoic acid, myristic acid, palmitic acid, stearic acid, oleic acid, linoleic acid, alpha-linolenic acid, and arachidic acid. The predominant fatty acids in HP oil are the unsaturated fatty acids: oleic acid (61.11%) and linoleic acid (25.84%). Among the saturated fatty acids, palmitic acid was the most abundant, accounting for 8.07%. The remaining six fatty acids collectively constituted less than 5% of the total composition. As indicated in Table 2, the high oleic acid content classifies HP oil within the group of oleic acid-rich oils. Notably, HP oil contains two dominant unsaturated fatty acids, oleic acid (ω-9) and linoleic acid (ω-6), respectively. A comparative analysis of the fatty acid profile of HP oil with other oilseeds suggests a similarity in both type and quantity of fatty acids to almond, pistachio, medium-oleic sunflower, and hazelnut oils [23].

Oleic acid is the most prevalent fatty acid in HP seed oil and holds significant importance for human health, playing a crucial role in the prevention of various diseases. This monounsaturated fatty acid is a primary component of phospholipids in the myelin membrane and is abundantly found in the phospholipid compounds of brain cell membranes. Research indicates a significant reduction in oleic acid levels in patients with Alzheimer’s disease and depression [40,41]. Furthermore, oleic acid contributes significantly to cardiovascular health and cholesterol management. Studies have shown that oleic acid can mitigate inflammatory parameters in the gut, reduce pathogenic bacteria such as *Escherichia coli*, and eliminate *Enterococcus faecalis* and *Candida glabrata*. Simultaneously, it promotes the restoration of beneficial bacteria like *Lactobacillus johnsonii* and *Bacteroides thetaiotaomicron*, thereby improving colitis. The combination of oleic acid with palmitic acid has also been shown to prevent the overgrowth of intestinal fungi and control inflammation by modulating the biodiversity of gut microbes [42]. Beyond its synergistic effects, oleic acid alone can control and improve intestinal inflammation, restore damaged intestinal epithelial barriers, and inhibit colitis [43]. Moreover, oleic acid significantly reduces oxidative stress and cellular inflammation [44], which can contribute to a reduction in systemic inflammation throughout the body. Other research also suggests that oleic acid can alleviate osteoarthritis symptoms [45].

### 3.3. Triacyclglycerols

Triacylglycerols (TAG), composed of glycerol and three attached fatty acids, are found in vegetable oils and human body tissue. They serve as a form of energy storage and release approximately twice as much energy during metabolism as carbohydrates and proteins [46]. In addition to their nutritional role, TAGs are also used in vegetable oils as a factor in determining the purity and recognition of oil and its adulterations. Each type of oil has a specific TAG profile that is unique to that oil, which imparts its unique nutritional, physical, and chemical characteristics. In other words, it expresses the qualitative characteristics of that oil [47].

In the study of HP seed oil, 14 types of TAGs were identified (Table 3). Of these, OOO, OOL, POO, and SOO were the predominant. OOO had the highest level, which is similar to high oleic sunflower oil (67–79%), olive oil (50–61%) and hazelnut oil (44–60%). The second most dominant TAG in HP seed oil was OOL (12.16%), which was similar to hazelnut oil (14–17%), and was also higher than sunflower oil (3–7%) and olive oil (7.3%). Also, POO (11.57%) in HP seed oil was in third place, which was lower than olive oil (22.96%) and similar to hazelnut oil (5–17%), but higher than sunflower oil. The fourth most dominant TAG was SOO (3.3%), which was similar to hazelnut oil (2.6%), and was lower than olive oil (7.07%) and sunflower (8.13%). Overall, HP seed oil is more similar to hazelnut oil in terms of TAG composition among the oils compared [47,48]. It is worth noting that due to the predominance of high amounts of oleic acid in its TAGs, it is very desirable and important from a nutritional point of view [49].

### 3.4. Phytosterols

Phytosterols are plant sterols structurally similar to cholesterol, and their analysis is crucial for detecting oil adulteration and identifying oils types. Beyond their role in quality control, phytosterols exhibit significant antioxidant properties, enhancing the thermal stability and oxidative shelf life of oils [50]. From a nutritional perspective, phytosterols are highly valued for their positive impact on human health. Vegetable oils are primary dietary sources of these compounds, which are known to reduce blood cholesterol levels, thereby lowering the risk of cardiovascular diseases. Furthermore, consumption of phytosterols, particularly those more abundant in unrefined oils, has been linked to the inhibition of cancer cell metastasis through the induction of apoptosis [50].

Seven distinct phytosterols were identified in HP oil (Table 4). Beta-sitosterol and stigmasterol were the most dominant compounds, accounting for 65.6% and 14%, respectively. The beta-sitosterol content in HP oil is comparable to that found in other widely consumed oils, such as sweet almond oil (73.0–86.0%), pistachio oil (75.0–94.0%), sunflower oil (56.0–58.0%), and hazelnut oil (76.54–96.0%) [23]. Beta-sitosterol, being the most abundant sterol in HP oil, plays a significant role in promoting heart health by reducing LDL cholesterol levels [51]. Moreover, it has demonstrated specific efficacy in managing benign prostatic hyperplasia (BPH) and controlling various inflammatory conditions [52].

### 3.5. Tocopherols

Tocopherols, commonly known as Vitamin E, are naturally synthesized in oilseeds, fruits, and actively dividing plant cells, primarily to prevent lipid oxidation [53]. This group comprises several forms, including α, β-, γ-, and δ-tocopherols. Among these, α- and γ-tocopherols are the most prevalent in food sources and accumulate significantly in human tissues, underscoring their critical biological importance [54].

In the analysis of HP seed oil, only α-tocopherol was identified among the four types of tocopherols (β, γ, α, δ). HP seed oil, with its exceptionally high α-tocopherol content, stands out as a rich natural source of this vitamin. Nutrition researchers emphasize the unique importance of α-tocopherol among other tocopherol isomers, often using “Vitamin E” synonymously with α-tocopherol due to its superior antioxidant and antitumor properties [55]. Furthermore, α-tocopherol, through its potent antioxidant and anti-inflammatory mechanisms, induces apoptosis in cancer cells by modulating the immune system. It also protects against cellular oxidative stress, neutralizes free radical reactions, and inhibits inflammatory and aging processes [56].

From the perspective of food industry experts, the presence of tocopherols in vegetable oils as natural antioxidants significantly extends their shelf life. For instance, adding only 0.2% α-tocopherol to oils such as olive, rapeseed, and palm oils, can enhances their thermal and oxidative stability. Beyond their preservative qualities, tocopherols also positively influence the aroma and flavor profiles of oils [57]. The α-tocopherol content in vegetable oils typically ranges between 200 and 1000 ppm [58]. Remarkably, this study found HP oil to be exceptionally rich in α-tocopherol (1610.90 ppm) (Table 5), a concentration that appears to be unprecedented when compared to many other common vegetable oils [23].

Specifically, the α-tocopherol content of HP seed oil significantly surpasses that of numerous other vegetable oils, including babassu oil (ND), coconut oil (ND–17 mg/kg oil), grape seed oil (ND–38 mg/kg oil), walnut oil (ND–170 mg/kg oil), palm oil (4–193 mg/kg oil), safflower seed oil (2–265 mg/kg oil), soybean oil (9–352 mg/kg oil), corn oil (23–573 mg/kg oil), rice bran oil (49–583 mg/kg oil), flaxseed oil (136–674 mg/kg oil), and even high-α-tocopherol sources like sunflower seed oil (403–935 mg/kg oil) [23].

### 3.6. Nutritional Quality Indices and Nutraceutical Properties

Oleic acid is the predominant fatty acid in HP seed oil and plays a vital role in human health. As the primary monounsaturated fatty acid in the bloodstream, it is instrumental in preventing various inflammatory diseases. Linoleic acid is the second most abundant fatty acid in HP oil. As a widely consumed monounsaturated fatty acid (MUFA), it serves as an important energy source and is an integral component of cell membrane structures, contributing to membrane fluidity [59,60]. While essential for health, an imbalance between linoleic acid and linolenic acid can raise health concerns [60,61]. To assess the nutritional quality of oils, researchers commonly utilize indices such as atherogenicity, thrombogenicity, and hypo-to-hypercholesterol ratios. These indices have been calculated for HP oil and other oils with similar fatty acid profiles, as presented in Table 6.

#### 3.6.1. Index of Atherogenicity (IA)

The term “atherogenic oils” refers to oils that contribute to atherosclerosis, commonly known as the hardening and narrowing of arteries. These oils typically contain high levels of saturated fatty acids or trans fats, which are known to elevate low-density lipoprotein (LDL) cholesterol (the “bad” cholesterol) while reducing high-density lipoprotein (HDL) cholesterol (the “good” cholesterol), thereby increasing the risk of cardiovascular events like heart attacks [62]. IA indicates the ratio of some saturated fatty acids to unsaturated fatty acids. A lower IA value is desirable, as it indicates a reduced likelihood of forming fatty plaques in arteries and a greater potential to decrease esterified fats and cholesterol [63].

The IA for HP seed oil was 0.13. This value is comparable to sunflower oil (0.13) and pistachio oil (0.14) but is slightly higher than almond oil (0.05) and hazelnut oil (0.09) [23]. When compared to cumin seed oil (0.46–0.53), HP oil shows a significantly lower IA. More notably, its IA is substantially lower than those of milk fat (1.42–5.13) and animal/meat fats (0.165–1.32) [64], suggesting a more favorable lipid profile regarding atherogenic risk. It should be noted that for milk fat, IA is in broad ranges, which can be due to the seasonal differences in the milk fat composition or animal type and nutrition.

#### 3.6.2. Index of Thrombogenicity (IT)

The Thrombogenicity Index (IT) assesses an oil’s potential to promote blood clot formation within blood vessels. Certain saturated fatty acids and trans fatty acids found in oils can elevate the risk of cardiovascular diseases by fostering clot formation. This index represents the ratio of palmitic, myristic, and stearic fatty acids to monounsaturated fatty acids [64]. Higher concentrations of palmitic, myristic, and particularly stearic saturated fatty acids are associated with an increased likelihood of blood clot formation and a higher IT value. Elevated levels of saturated fatty acids are directly linked to an increased risk of heart disease mortality [65]. In essence, a lower IT value indicates a reduced potential for thrombosis and signifies a healthier oil. The thrombogenicity index of HP oil was 0.24. The IT values for various fats include meat fat (0.28–1.69) [66], and vegetable seed fat (0.136–0.56) [67]. The IT of HP oil is lower than that of dairy fats [65], pistachio oil (0.31) and hazelnut oil (0.19), comparable to sunflower oil (0.25), but slightly higher than almond oil (0.14) when compared to oils with similar fatty acid profiles [28]. This suggests that HP oil has a relatively low potential for promoting blood clot formation.

#### 3.6.3. Hypo/Hyper Index

Elevated cholesterol concentrations are a major risk factor for cardiovascular diseases, and recent research highlights a clear relationship between fatty acid profiles and blood cholesterol levels and types. Palmitic (C16:0) and stearic (C18:0) fatty acids, as saturated fatty acids, are directly implicated in increasing the risk of cardiovascular diseases by raising levels of “bad” cholesterol. Conversely, unsaturated fatty acids like alpha-linolenic acid (an omega-3 fatty acid) exert positive effects on cardiovascular health. For instance, consuming food sources rich in alpha-linolenic acid, such as fish (at a daily intake of 45 g), has been shown to reduce the risk of heart disease by 60% [68].

The Hypo/Hyper Index evaluates an oil’s impact on the ratio of “bad” cholesterol (LDL) to “good” cholesterol (HDL). A lower value for this index indicates that the oil is less likely to produce LDL and more likely to promote HDL [64]. For example, olive oil, rich in monounsaturated fatty acids like oleic acid, can decrease LDL and subsequently increase HDL [68]. In contrast, oils abundant in saturated fatty acids like palmitic and stearic acids, such as palm and coconut oils, elevate LDL levels and are directly associated with ischemic heart and brain diseases [69].

In this study, the Hypo/Hyper Index for HP seed oil was 9.77. This value is lower than almond oil (20.71) and hazelnut oil (13.15), slightly higher than pistachio oil (7.55), and similar to sunflower oil (9.20) (Table 6). These findings suggest that HP oil possesses a favorable Hypo/Hyper Index, implying a positive impact on cholesterol balance.

## 4. Conclusions

The findings of this study confirm that *Heracleum persicum* (HP) seeds represent a promising source of vegetable oil. The fatty acid profile of HP oil, characterized by the predominance of oleic acid and linoleic acid, comparable to well-known healthy oils such as almond, pistachio, sunflower, and hazelnut oils. Furthermore, HP oil is rich in valuable bioactive compounds, including phenolic compounds, and contains significant amounts of phytosterols such as beta-sitosterol and stigmasterol. Notably, its exceptionally high content of alpha-tocopherol, positions it as a rich source of natural Vitamin E. Also, when assessed using nutritional indices such as the atherogenicity index, thrombogenicity index, and the hypocholesterolemic to hypercholesterolemic ratio, HP oil demonstrates a favorable profile compared to oils with similar fatty acid compositions. While these results are highly encouraging, further researches are suggested to explore its potential applications in food, cosmetics and pharmaceutical industries, and to facilitate its successful commercialization.

## Figures and Tables

**Figure 1 foods-14-03486-f001:**
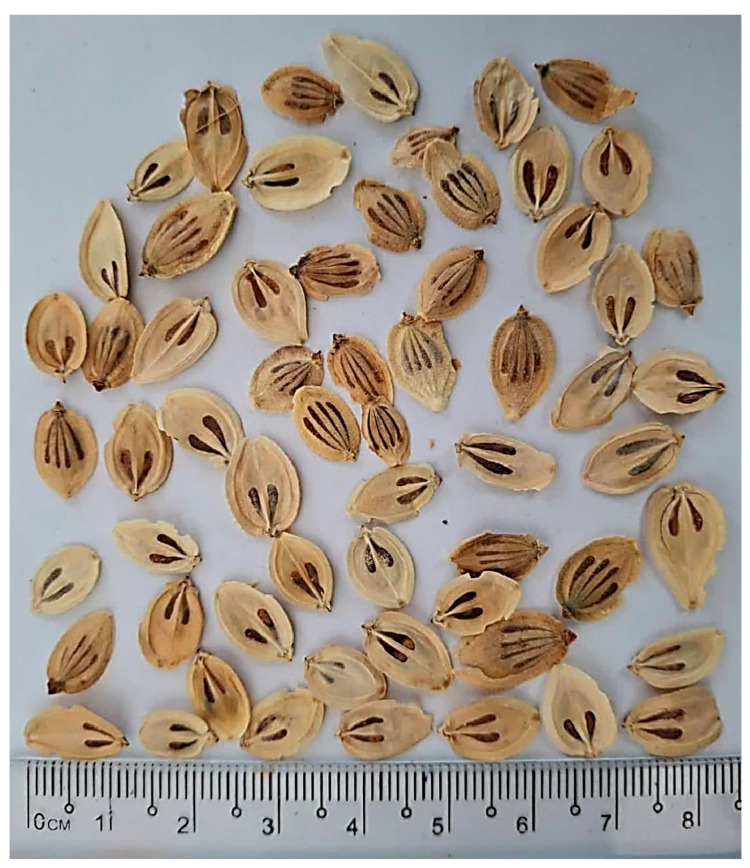
Seeds of *Heracleum persicum*.

**Table 1 foods-14-03486-t001:** Physical and chemical characteristics of *Heracleum persicum* seed and its oil.

Characteristics	Results
**Seed**	
Oil percent (%)	12.6 ± 0.2 *
Moisture (%)	6.67 ± 0.1
**Extracted oil**	
Specific gravity	0.876 ± 0.0
Refractive index	1.464 ± 0.0
Acid value (mg KOH/g oil)	3.86 ± 0.1
Peroxide value (mEq O_2_/kg oil)	5.95 ± 0.2
Carotenoid content (mg/kg oil)	4.95 ± 0.3
Chlorophyll content (mg/kg oil)	28.34 ± 0.1
Total phenolic compounds (mg/kg oil)	4.1 ± 0.4

* Mean is shown as ±SD.

**Table 2 foods-14-03486-t002:** Comparison of unsaturated and saturated fatty acids in *Heracleum persicum* (HP) seeds oil with other similar vegetable oils.

Fatty Acid	HP Seeds Oil	Almond Oil [23]	Pistachio Oil [23]	Sunflower [23] (Mid-Oleic Acid)	Hazelnut Oil [23]
Dodecanoic acid	1.11 ± 0.02 *	ND **	ND	ND	ND
Myristic acid	0.88 ± 0.01	ND–0.1	ND–0.6	ND–1	ND–0.1
Palmitic acid	8.07 ± 0.32	4.0–9.0	8.0–13.0	4.0–5.5	4.2–8.9
Palmitoleic acid	0.37 ± 0.14	0.2–0.8	ND–0.2	ND–0.05	ND–0.5
Stearic acid	1.53 ± 0.11	ND–3.0	0.5–0.35	2.1–5.0	0.8–3.2
Oleic acid (cis)	61.11 ± 0.16	62.0–76.0	50.0–70.0	43.1–71.8	74.2–86.7
Linoleic acid (cis)	25.84 ± 0.22	20.0–30.0	8.0–34.0	18.7–45.3	5.2–18.7
α-linolenic acid	0.57 ± 0.01	ND–0.5	0.1–1.0	ND–0.5	ND–0.6
Arachidic acid	0.18 ± 0.06	ND–0.5	ND–0.3	0.2–0.4	ND–0.3

* Mean is shown as ±SD. ** ND = not detectable.

**Table 3 foods-14-03486-t003:** Comparison of triacylglycerol compositions of HP seeds oil seed oil with other similar vegetable oils (expressed as percentage).

TAG	HP Seeds Oil	Sunflower [47] (Mid-Oleic Acid)	Hazelnut Oil [47]	Olive Oil [48]
LLL	0.73	ND *	ND–2	0.01
OLL	1.35	ND	2.91	0.04
OLLn	0.17	ND	ND	0.13
PLL	0.73	ND	ND–1.5	1.31
OOLn	ND	ND	ND–0.5	ND
OOL	12.16	3–7	14–17	7.37
POL	1.68	ND	1–5	2.97
PLO	ND	ND	ND	ND
PPL	0.43	1–4	0.3–0.6	ND
OOO	61.11	67–79	44–60	50.61
POO	11.57	ND	5–17	22.96
PPO	0.22	3.5–7.3	ND–1.7	2.81
SOO	3.3	8–13	2–6	7.07
SOS	0.26	ND–1	ND	ND
PSO	ND	ND–1	0.5–1.5	ND
POS	0.41	ND	ND	1.44
Total	94.12	~96.90	~96.21	97.12

TAG: Triacylglycerol, O: Oleic acid, L: Linoleic acid, Ln: Alpha-linolenic acid, P: Palmitic acid, S: Stearic acid. * ND = not detectable.

**Table 4 foods-14-03486-t004:** Phytosterol composition of *Heracleum persicum* (HP) seed oil in comparison with other vegetable oils.

Phytosterol	HP	Almond [23]	Pistachio [23]	Sunflower [23]	Hazelnut [23]
Cholesterol	0.4 ± 0.1 *	ND **–0.1	ND–1.0	0.1–0.2	ND–1.1
Campesterol	3.6 ± 0.2	5.0–2.0	4.0–6.5	9.1–9.6	3.0–6.2
Stigmasterol	14.0 ± 2.6	4.0–0.4	0.5–7.5	9.0–9.3	ND–2.0
Betasitosterol	65.6 ± 3.3	73.0–86.0	75.0–94.0	56.0–58.0	76.54–96.0
Delta5-avenasterol	3.2 ± 0.4	5.0–14.0	6.0–8.0	4.8–5.3	1.0–5.1
Delta7-stigmastanol	3.2 ± 0.3	ND–3.0	ND–0.7	7.7–7.9	ND–4.3
Delta7-avenasterol	1.7 ± 0.1	ND–3.0	ND–0.5	4.3–4.4	ND–1.6
Total (mg/kg oil)	3266.67	1590–4590	1840–5400	1700–5200	1200–1800

* Mean is shown as ±SD. ** ND = not detectable.

**Table 5 foods-14-03486-t005:** Comparing the tocopherol content of HP seed oil with other vegetable oils.

Tocopherol (mg/kg Oil)	HP	Almond [23]	Pistachio [23]	Sunflower [23]	Hazelnut [23]
α-tocopherol	1610.90 ± 13.2 *	20–545	10–330	403–935	100–420
β-tocopherol	ND **	10–ND	ND	ND–45	6–12
γ-tocopherol	ND	104–ND	0–100	ND–34	18–194
δ-tocopherol	ND	ND–5	ND–50	ND–0.7	ND–10

* Mean is shown as ±SD. ** ND = not detectable.

**Table 6 foods-14-03486-t006:** Nutritional quality indices (NQI) of *Heracleum persicum* (HP) oil and its comparison with other similar oils in terms of fatty acid profile.

NQI	HP Seeds Oil	Almond Oil	Pistachio Oil	Sunflower	Hazelnut Oil
Atherogenicity	0.13	0.05	0.14	0.13	0.09
Thrombogenicity	0.24	0.14	0.31	0.25	0.19
Hypo/Hyper *	9.77	20.71	7.55	9.20	13.15

* Hypocholesterolemic to Hypercholesterolemic.

## Data Availability

Detailed data supporting the findings of this study are available from the corresponding authors upon reasonable request.

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
