# Peer review of "Extraction of Seed Oil from Heracleum persicum Desf. ex Fischer and Investigation of Its Composition, Qualitative and Nutraceutical Properties"

_foods, 2025, doi:10.3390/foods14203486_

Round 1
Reviewer 1 Report
Comments and Suggestions for Authors
The paper of Dadazadeh et al. “Extraction of Seed Oil from Heracleum persicum Desf.…” aimed to chemical study new vegetable oil from Heracleum persicum seed. Briefly, the manuscript presents the results of a very simple experiment, which is not enough for a full research publication and can be considered only as a short communication.
Highlights and strengths of the manuscript are:
New data about Heracleum persicum seed oil expand knowledge about new vegetable oils.
Specific comments and suggested revisions:
- I would like to get an answer to the question about the applicability of Heracleum persicum seed oil as a vegetable oil, provided that it has a pronounced smell. The main components of Heracleum persicum seed oil are hexyl butyrate (20.9–44.7%), octyl acetate (11.2–20.3%), hexyl-2-methylbutyrate (4.81–8.64%), and octyl 2-methyl butyrate (3.41–8.91%) [https://doi.org/10.3390/molecules27196296] or in the early development of seeds stage, hexyl butyrate (32.1%), and octyl acetate (11.7%), in the mid‐mature seeds stage hexyl butyrate (38.8%), octyl acetate (14.5%), in the late‐mature/ripe seeds stage, hexyl butyrate (23.6%), and octyl acetate (10.5%) [https://onlinelibrary.wiley.com/doi/10.1002/fsn3.1916]. Two main compounds, hexyl butyrate and octyl acetate, are volatiles with strong odor [https://www.thegoodscentscompany.com]. The content of essential oil in the seeds is up to 3.5%, which means that in vegetable oil it will be more than 20%. Vegetable oils have a smell, but it is very weak. In this regard, the concept of using Heracleum persicum seed oil as a vegetable oil seems questionable (but why not... tastes differ).
- 2.2. “… hexane solvent was added in a ratio of 1 to 10”. Is that means that “… hexane solvent was added in a seed : hexane ratio of 1:10”?
- Table 1. Physical and chemical characteristics of Heracleum persicum seed oil. But “Oil percent” (percentage?) and “Moisture” are not Heracleum persicum seed oil characteristics (only Heracleum persicum seeds). Please correct.
- From tables 2, 3, 5 it is not clear how the data for other oils were obtained. Judging by how it is presented in the manuscript, you realized research on four more oils. But in the materials and methods section there is no information of other oils.
- Table 4 is not necessary as it contains only one significant value. You can insert this value into the text.
Overall, the amount of experimental data presented is insufficient for publication in a Q1 journal. All the material in the manuscript can easily fit into one table. The article does not offer a solution to any important scientific problem. It is simply data on the composition of Heracleum persicum seed oil. With all due respect, this doesn't look like a complete study. Authors should clearly formulate the purpose of the work, add correct and reliable data and a clear discussion.
Author Response
Comments and Suggestions for Authors
The paper of Dadazadeh et al. “Extraction of Seed Oil from Heracleum persicum Desf.…” aimed to chemical study new vegetable oil from Heracleum persicum seed. Briefly, the manuscript presents the results of a very simple experiment, which is not enough for a full research publication and can be considered only as a short communication.
Highlights and strengths of the manuscript are:
New data about Heracleum persicum seed oil expand knowledge about new vegetable oils.
Specific comments and suggested revisions:
Cooment 1. I would like to get an answer to the question about the applicability of Heracleum persicum seed oil as a vegetable oil, provided that it has a pronounced smell. The main components of Heracleum persicum seed oil are hexyl butyrate (20.9–44.7%), octyl acetate (11.2–20.3%), hexyl-2-methylbutyrate (4.81–8.64%), and octyl 2-methyl butyrate (3.41–8.91%) [https://doi.org/10.3390/molecules27196296] or in the early development of seeds stage, hexyl butyrate (32.1%), and octyl acetate (11.7%), in the mid‐mature seeds stage hexyl butyrate (38.8%), octyl acetate (14.5%), in the late‐mature/ripe seeds stage, hexyl butyrate (23.6%), and octyl acetate (10.5%) [https://onlinelibrary.wiley.com/doi/10.1002/fsn3.1916]. Two main compounds, hexyl butyrate and octyl acetate, are volatiles with strong odor [https://www.thegoodscentscompany.com]. The content of essential oil in the seeds is up to 3.5%, which means that in vegetable oil it will be more than 20%. Vegetable oils have a smell, but it is very weak. In this regard, the concept of using Heracleum persicum seed oil as a vegetable oil seems questionable (but why not... tastes differ).
Response 1:
Authors thank you very much for your valuable comments and suggestion, which all were included in the revised manuscript as advised.
Exactly, Heracleum persicum has a specific odor due to its particular essential oils, but the oil extracted from this seed may have a strong anti-inflammatory effect due to its secondary metabolites and its richness in other bioactive components such as tocopherols and also it can have many positive health effects due to its suitable fatty acid composition as well. To overcome its unpleasant odor, for medicinal uses, techniques such as filling in soft gels or enteric-coated capsules can be suitable options. This research is just a preliminary study to introduce this oil for further researches on its nutritional, pharmaceutical or cosmetics applications.
Comment 2. 2.2. “… hexane solvent was added in a ratio of 1 to 10”. Is that means that “… hexane solvent was added in a seed: hexane ratio of 1:10”?
Response 2: This was clarified in the text. n-hexane was used to extract the oil. Solvent to seed powder was 10 to 1 (v/w).
Comment 3. Table 1. Physical and chemical characteristics of Heracleum persicum seed oil. But “Oil percent” (percentage?) and “Moisture” are not Heracleum persicum seed oil characteristics (only Heracleum persicum seeds). Please correct.
Response 3: It was revised as advised.
Comment 4. From tables 2, 3, 5 it is not clear how the data for other oils were obtained. Judging by how it is presented in the manuscript, you realized research on four more oils. But in the materials and methods section there is no information of other oils.
Response 4:
Sorry for this error. These data are obtained from the standard tables of the Codex Elementaries, for which we have cited their sources in the table.
Comment 5. CTable 4 is not necessary as it contains only one significant value. You can insert this value into the text.
Response 5:
We expanded Table 4 as other Tables and included other data for comparison.
Comment 6. Overall, the amount of experimental data presented is insufficient for publication in a Q1 journal. All the material in the manuscript can easily fit into one table. The article does not offer a solution to any important scientific problem. It is simply data on the composition of Heracleum persicum seed oil. With all due respect, this doesn't look like a complete study. Authors should clearly formulate the purpose of the work, add correct and reliable data and a clear discussion.
Response 6:
Authors agree with the comment of the esteemed reviewer; however, more analysis were performed and their data were included in addition to extensive revision according to your valuable comments and suggestions to improve and strength the manuscript results to meet journal requirements.
Reviewer 2 Report
Comments and Suggestions for Authors
General comment
This study systematically evaluates the nutritional quality and nutraceutical properties of Heracleum persicum seed oil, focusing on fatty acid composition, atherogenicity, thrombogenicity, and hypo/hypercholesterolemic indices. The work highlights oleic acid as the dominant fatty acid and discusses its implications for cardiovascular, neurological, and intestinal health. By comparing HP oil with other common edible oils, the manuscript provides valuable insights into its potential health benefits and suitability as a functional oil. The comprehensive presentation of nutritional quality indices offers a useful reference for both researchers and food industry applications. However, certain aspects require clarification, including the consistency of comparative data sources, statistical significance of index differences, and justification of health-related interpretations.
Line 33-62 While the introduction provides detailed botanical characteristics and some pharmacological findings of Heracleum persicum, it devotes excessive space to morphological descriptions. The background information directly related to the seed oil—such as extraction methods, nutritional functions, and comparison with existing oil sources—is insufficient. I recommend reducing morphological details and expanding the literature review on seed oil extraction, composition, and nutritional relevance to better highlight the scientific significance and novelty of the study.
Line 63-162 Several analytical methods are cited only generally, and key experimental conditions (e.g., sample size, replication number, temperature range, standards used) are missing for some measurements such as moisture, pigments, fatty acids, phytosterols, and tocopherols. I recommend including these essential parameters to ensure methodological reproducibility.
Line 177-184 In Section 3.1.2, the specific gravity of HP oil is compared with various vegetable oils, but the discussion lacks interpretation of the practical implications for oil identification and fraud detection. For example, does the slightly higher value compared to palm oil indicate any compositional or processing particularities? I recommend that the authors provide an analysis of the potential chemical or physical reasons behind these differences and explore their relevance in industrial or food quality control applications.
Line 220-227 In Section 3.1.6, the carotenoid content of HP oil is reported to be considerably lower than that of many other vegetable oils. However, the discussion only presents numerical comparisons without exploring possible causes (e.g., varietal differences, maturity, extraction process, or storage conditions). Moreover, there is no analysis of how the low carotenoid content might affect shelf life or antioxidant capacity. I recommend adding discussion on the possible reasons for this low content and its potential implications.
Line 236-244 In Section 3.1.8, the phenolic content of HP oil is reported as 1.4 mg/kg, which is relatively low compared to other oils. However, the authors do not address the sensitivity of the detection method, the sample preparation procedure, or potential sources of loss, all of which could affect comparability with literature data. I recommend providing methodological details in relation to previous studies and discussing how low phenolic content may influence antioxidant performance and nutritional value.
Line 291-301 The α-tocopherol content is described as “unprecedented” compared to other vegetable oils; however, no statistical comparison or literature range table is provided to substantiate this claim. The authors should support this statement with comparative data and appropriate references.
Line 325-335 The IA comparison relies heavily on literature values for other oils, but the sources report broad ranges (e.g., milk fat 5.13–1.42), which may reflect differences in analytical conditions or sample origins. The authors should clarify whether all values were recalculated using a consistent formula from the same dataset, or discuss the potential limitations of comparing data from heterogeneous studies.
Line 351-371 The discussion of the Hypo/Hyper Index is largely qualitative, stating that HP oil has a “good” value, but does not define a threshold for what is considered favorable in health terms. The authors should justify the criteria for categorizing the value as beneficial and, if possible, relate it to clinical outcomes or dietary recommendations.
Author Response
Comment 1. This study systematically evaluates the nutritional quality and nutraceutical properties of Heracleum persicum seed oil, focusing on fatty acid composition, atherogenicity, thrombogenicity, and hypo/hypercholesterolemic indices. The work highlights oleic acid as the dominant fatty acid and discusses its implications for cardiovascular, neurological, and intestinal health. By comparing HP oil with other common edible oils, the manuscript provides valuable insights into its potential health benefits and suitability as a functional oil. The comprehensive presentation of nutritional quality indices offers a useful reference for both researchers and food industry applications. However, certain aspects require clarification, including the consistency of comparative data sources, statistical significance of index differences, and justification of health-related interpretations.
Response 1:
Authors would like to thank you very much for your positive and encouraging comments and the manuscript was extensively revised as advised.
Comment 2. Line 33-62 While the introduction provides detailed botanical characteristics and some pharmacological findings of Heracleum persicum, it devotes excessive space to morphological descriptions. The background information directly related to the seed oil—such as extraction methods, nutritional functions, and comparison with existing oil sources—is insufficient. I recommend reducing morphological details and expanding the literature review on seed oil extraction, composition, and nutritional relevance to better highlight the scientific significance and novelty of the study.
Response 2:
The nutritional importance of angelica seed oil was further discussed with respect to its compounds such as the fatty acid oleic acid and alpha-tocopherol. Introduction section was also revised as advised.
Comment 3. Line 63-162 Several analytical methods are cited only generally, and key experimental conditions (e.g., sample size, replication number, temperature range, standards used) are missing for some measurements such as moisture, pigments, fatty acids, phytosterols, and tocopherols. I recommend including these essential parameters to ensure methodological reproducibility.
Response 3:
The detail parameters were included in the experiments as advised.
Comment 4. Line 177-184 In Section 3.1.2, the specific gravity of HP oil is compared with various vegetable oils, but the discussion lacks interpretation of the practical implications for oil identification and fraud detection. For example, does the slightly higher value compared to palm oil indicate any compositional or processing particularities? I recommend that the authors provide an analysis of the potential chemical or physical reasons behind these differences and explore their relevance in industrial or food quality control applications.
Response 4:
Measuring specific gravity and comparing it with the specific gravity of other oils can only be used as an indicator to indicate the physical similarity or difference of the oil, which, with the help of other physical and chemical characteristics, can help identify, distinguish and purity of oils, which were explained in the text as advised.
Comment 5. Line 220-227 In Section 3.1.6, the carotenoid content of HP oil is reported to be considerably lower than that of many other vegetable oils. However, the discussion only presents numerical comparisons without exploring possible causes (e.g., varietal differences, maturity, extraction process, or storage conditions). Moreover, there is no analysis of how the low carotenoid content might affect shelf life or antioxidant capacity. I recommend adding discussion on the possible reasons for this low content and its potential implications.
Response 5:
It was explained in the discussion section as advised.
Comment 6. Line 236-244 In Section 3.1.8, the phenolic content of HP oil is reported as 1.4 mg/kg, which is relatively low compared to other oils. However, the authors do not address the sensitivity of the detection method, the sample preparation procedure, or potential sources of loss, all of which could affect comparability with literature data. I recommend providing methodological details in relation to previous studies and discussing how low phenolic content may influence antioxidant performance and nutritional value.
Response 6:
It was discussed in detail in the discussion section as advised.
Comment 7. Line 291-301 The α-tocopherol content is described as “unprecedented” compared to other vegetable oils; however, no statistical comparison or literature range table is provided to substantiate this claim. The authors should support this statement with comparative data and appropriate references.
Response 7:
A numerical comparison of the tocopherol content of HP oil with other oils was performed according to the Codex Alimentarius standard as recommended.
Comment 8. Line 325-335 The IA comparison relies heavily on literature values for other oils, but the sources report broad ranges (e.g., milk fat 5.13–1.42), which may reflect differences in analytical conditions or sample origins. The authors should clarify whether all values were recalculated using a consistent formula from the same dataset, or discuss the potential limitations of comparing data from heterogeneous studies.
Response 8:
IA was calculated according to the formula presented in the manuscript for vegetable oils, but it was reported from the previously published data for milk fat, which is in broad ranges as esteemed reviewer also mentioned. It can be due to the seasonal differences in the milk fat composition or animal type and nutrition as well which were explained in text as advised..
Comment 9. Line 351-371 The discussion of the Hypo/Hyper Index is largely qualitative, stating that HP oil has a “good” value, but does not define a threshold for what is considered favorable in health terms. The authors should justify the criteria for categorizing the value as beneficial and, if possible, relate it to clinical outcomes or dietary recommendations.
Response 9:
Authors agree with this comment, but unfortunately, there is no agreed-upon standard number for the normal range (good or bad) of this index in oils, and oils are only compared relatively through this index, with the lower the number, the more beneficial it is for health.
Round 2
Reviewer 1 Report
Comments and Suggestions for Authors
After correcting the comments, the manuscript looks much better. I can say that I have no other comments.
Author Response
Dear Reviewer,
We sincerely thank the reviewer for their positive feedback and for recognizing the improvements made in the revised manuscript. We appreciate the time and effort dedicated to reviewing our work and are glad that all concerns have been addressed satisfactorily.
Thank you very much again.
With best regards,
Sodeif Azadmard-Damirchi,